# MicroRNAs (miRNAs) in Cardiovascular Complications of Rheumatoid Arthritis (RA): What Is New?

**DOI:** 10.3390/ijms23095254

**Published:** 2022-05-08

**Authors:** Daniela Maria Tanase, Evelina Maria Gosav, Daniela Petrov, Dan-Stefan Teodorescu, Oana Nicoleta Buliga-Finis, Anca Ouatu, Ionut Tudorancea, Elena Rezus, Ciprian Rezus

**Affiliations:** 1Department of Internal Medicine, “Grigore T. Popa” University of Medicine and Pharmacy, 700115 Iasi, Romania; tanasedm@gmail.com (D.M.T.); dan-stefan.teodorescu@d.umfiasi.ro (D.-S.T.); oana_finish@yahoo.com (O.N.B.-F.); ank_mihailescu@yahoo.com (A.O.); ciprianrezus@yahoo.com (C.R.); 2Internal Medicine Clinic, “St. Spiridon” County Clinical Emergency Hospital Iasi, 700111 Iasi, Romania; 3Department of Rheumatology and Physiotherapy, “Grigore T. Popa” University of Medicine and Pharmacy, Iasi 700115, Romania; danielapetrovdoc@gmail.com (D.P.); elena_rezus@yahoo.com (E.R.); 4I Rheumatology Clinic, Clinical Rehabilitation Hospital, 700661 Iasi, Romania; 5Department of Morpho-Functional Sciences II, Discipline of Physiology, “Grigore T. Popa” University of Medicine and Pharmacy, 700115 Iasi, Romania; ionut.tudorancea@umfiasi.ro; 6Cardiology Clinic, “St. Spiridon” County Clinical Emergency Hospital, 700111 Iasi, Romania

**Keywords:** microRNAs, miRNAs, rheumatoid arthritis, RA, cardiovascular complications, CVD, atherosclerosis, pericarditis, myocardial infarction

## Abstract

Rheumatoid Arthritis (RA) is among the most prevalent and impactful rheumatologic chronic autoimmune diseases (AIDs) worldwide. Within a framework that recognizes both immunological activation and inflammatory pathways, the exact cause of RA remains unclear. It seems however, that RA is initiated by a combination between genetic susceptibility, and environmental triggers, which result in an auto-perpetuating process. The subsequently, systemic inflammation associated with RA is linked with a variety of extra-articular comorbidities, including cardiovascular disease (CVD), resulting in increased mortality and morbidity. Hitherto, vast evidence demonstrated the key role of non-coding RNAs such as microRNAs (miRNAs) in RA, and in RA-CVD related complications. In this descriptive review, we aim to highlight the specific role of miRNAs in autoimmune processes, explicitly on their regulatory roles in the pathogenesis of RA, and its CV consequences, their main role as novel biomarkers, and their possible role as therapeutic targets.

## 1. Introduction

Rheumatoid Arthritis (RA) is a chronic autoimmune disease (AID), characterized by chronic systemic inflammation that occurs especially in joint synovitis [1]. With heterogeneous pathophysiology, it is an immune disorder characterized by the presence or the absence of autoantibodies (seropositivity of anti-citrullinated peptide antibodies (ACPAs) and of rheumatoid factor (RF) antibodies) [2]. The worldwide incidence of RA is about 0.25%, with a higher prevalence in urban than in rural areas [3]. According to statistics, the risk of developing this disease is around 3.6% in women compared to a 1.7% risk in men [4]. RA has a considerable individual and social burden, and discovering new biomarkers that would allow the establishment of an early diagnosis, and potentially prevent the installation of cardiovascular complications, are eagerly desired [3,5].

All autoimmune diseases are characterized by a complicated immunity disbalance, which leads to a loss of self-tolerance following an attack on endogenous tissues and cells. This irregular immune response in the synovial tissues is usually generated by complex interactions between genetic background and environmental factors [6,7]. RA can lead to the onset of different extraarticular manifestations, including the development of cardiovascular diseases (CVD) [8,9,10]. Also, patients with RA are also, at increased risk of acquiring infections such as Coronavirus Disease 2019 (COVID-19) [11]. As COVID-19 became a worldwide health confrontation, with an increased rate of death, patients with immunologic dysfunction are more susceptible to develop complications regardless of ongoing treatment [12]. Scientific data array that altered interrelated post-transcriptional and epigenetic mechanisms, such as histone modifications, DNA methylation changes, and microRNA (miRNAs) activity variations, act jointly by altering gene and protein expression levels, and contribute to AID and different CV disorders [13,14]. Many studies have pointed out the important role of miRNAs in RA [15,16,17,18], and in CVD diseases onset and development [19,20]. In vivo research described miRNAs as having potential predictive and/or prognostic roles as biomarkers for the evaluation of treatment response and/or disease activity [21,22]. Considering AID continues to remain a challenge for clinicians, over the last decade miRNAs research has grown substantially, their therapeutic potential also being of great interest [23,24]. 

In this review, we aim to highlight the specific role of miRNAs in autoimmune processes, explicitly on their regulatory roles in the pathogenesis of rheumatoid arthritis, and its CV consequences, their main role as novel biomarkers, and their possible role as therapeutic targets. 

## 2. MicroRNAs

MicroRNAs are small Non-coding RNAs (ncRNAs) (18–25 nucleotides long), with an important role in innate and adaptative immunity, and in regulating gene expression without alteration of the nucleotide sequence at the post-transcriptional stage [25]. In the human body there are among 200 and 255 genes that encode miRNAs [26], transcribed from DNA sequences into primary miRNAs (pri-miRNAs) and converted into precursor; miRNAs (pre-miRNAs) and mature miRNAs [27]. There are two forms of miRNA biogenesis, the canonical and non-canonical pathway [28]. This class of nonprotein-coding RNAs consists of housekeeping RNAs: ribosomal ribonucleic acid (rRNAs), small nuclear RNA (snRNAs), small nucleolar RNAs (snoRNAs), transfer RNA (tRNAs), and regulatory RNAs that include short non-coding RNA (ncRNAs) < 200 nucleotides, microRNA (miRNAs), piwi-interacting RNA (piRNAs) and longer than 200 nucleotides (lncRNAs) [29].

MiRNAs are not only involved in important cellular processes, but also, their aberrant expression has been described in many chronic and acute diseases [27,28].

### 2.1. Role of MiRNAs in RA

As one of the regulators of gene expression, miRNAs can influence immune homeostasis, immune cell development, differentiation, proliferation, and unevenness of proinflammatory/anti-inflammatory cytokines [29]. The pathogenesis behind RA remains unclear, but there are ongoing scientific investigations which aim not only to elucidate the all the mechanism, but also, to provide novel detection and treatment options for RA [30].

Recent reports are showing that epigenetic mechanisms via miRNAs, play a significant role in RA pathogenesis [31]. Interestingly, some miRNAs can serve as potential biomarkers for detecting immune diseases such as RA, and improve CVD risk prediction models [13]. For example, miR-16, miR-125b, and miR-223 can serve as markers of therapeutic response for conventional disease-modifying antirheumatic drugs (DMARDs), while miR-22, miR-23, miR-27a, miR-125b, miR-223, and miR-886 can be used in biologic DMARDs [32]. Furthermore, certain levels of circulating miRNAs help to predict early RA compared to long-standing disease [17]. Filková et al. [33] described that there are lower levels of miR-146 in the early stages of RA than in the established disease [33]. Serum levels of miR-486, miR-38 especially miR-22, were elevated in those who finally developed RA [32]. Of note, the miR-22 levels are different in patients with RF and without (*p* = 0.04), and they are positively correlated with the erythrocyte sedimentation rate (ESR), C-reactive protein (CRP), and disease activity score (DAS28) which makes it a potential molecular marker for RA activity and exacerbation [2,34]. Additionally, miR 22, and miR-451 in T cells were linked with elevated DAS28, ESR, and Interleukin-6 (IL-6) [35], miR-125a with a high CRP level [36], and miR-125b expression with ESR, CRP, and DAS28 levels [32]. Contrarily, a recent study reported a negative correlation between markers of inflammatory conditions, ESR, and the expression of miRNAs in male patients [37].

Since inflammatory cytokines play an important role in the pathogenesis of RA, perhaps specific miRNAs may decrease the inflammatory cytokine levels in RA. On this subject, miR-137 has been known as a regulator of susceptibility genes in RA. Once it is overexpressed the level of pro-inflammatory markers like IL-1b, IL-6, and cyclooxygenase-2 (COX2) decreases [38]. Hence, some of the crucial roles of miRNAs in RA are displayed by disease activity, therapeutic response, and susceptibility to developing cardiovascular complications based on the activity of the disease [39]. According to Singh et al. [40], blood samples of methotrexate (MTX) treated subjects’ responders, indicated lower baseline levels of miR-132, miR-146a, and miR-155 than in non-responders [40]. Growing data shows that are some miRNAs that serve as a prediction tool of therapeutic effectiveness. In their research Cunnigham and coworkers reported lower levels of miR-339-5p and let-7i-5p (*p* < 0.01), following methotrexate (MTX) treatment [41]. Similarly, the miR-55 reduction could expose the effects of methylprednisolone (MP) on CD4+ T cells. Once MP decreases the level of miR55, the expression of the suppressor of cytokine signaling (SOCS)1 increases and with that, the amount of Janus kinase (JAK)/signal transducer and activator transcription (STAT) signaling declines, thereby showing the double effect of miRNA-55, as a marker of therapeutic effectiveness and its potential as a therapeutic target [28,42]. 

The development of biotechnology has provided a gamut of biological treatments, but the possibility of developing different side effects required the presence of some markers for the prediction of disease outcomes [43]. MiR-125b [44], miRNA-146a [45], miR-431-5p [46], miRNA-125a-p [47], are some of the miRNAs useful in the prediction of biological treatment. In an excellent revision by Duroux-Richard et al. [28,43], which included 48 RA patients, 32 of them who were treated with rituximab showed disclosed deregulated miR-125b in blood and serum samples [43]. Results showed that high expression of miR-125b was associated with a good response to anti-CD20 therapy. Patients with RA with low miR-125b expression at the time of onset of the disease are significantly less likely to improve clinically after 3 months of rituximab treatment. Serum miR-125b abundance may be used as a biomarker for treatment prediction [43,44]. On the contrary, miR-431-5p was reduced in human RA fibroblast-like synoviocytes (FLS) cells with tumor necrosis factor (TNF-α) treatment, compared with those without TNF-α treatment (*p* = 0.001). Once the expression of miR-431-5p decreased, the initial end phase apoptosis in HFLS-RA cells (*p* < 0.0001) was repressed [46]. These findings suggest that miRNAs such as miR-431-5p may have value as novel predictive and prognostic biomarkers in the evolution and treatment of RA.

Furthermore, the differential expression of miRNAs, emphasizes the polymorphism of therapeutic effects in RA patients. Establishing a premature diagnosis is vital to avoid possible cardiovascular RA consequences and future disability [2]. The existence of different markers may improve distinguishing between healthy controls and those predisposed to develop RA [48]. Cunnigham et.al [41] in a complex study, found eight miRNAs of interest (miR-126-3p, miR-221-3p, miR-24-3p, miR-130a-3p, miR- 339-5p, let-7i-5p let-7d-5p, miR-431-3p,) that were increased in the RA group comparatively with the control group (all *p* ≤ 0.01), and showed their potential role as biomarkers of RA development. Others pointed out that miR-126-3p with an area under the curve (AUC) of 0.8724 (*p* < 0.0001), miR-221-3p, and let-7d-5p (*p* ≤ 0.0001), were the most sensitive and specific miRNAs and have the highest diagnostic potential for RA [41]. This data indicates that there are other miRNA profiles in the serum of RA patients compared to healthy ones. Also, the serum of patients with arthralgia (joint pain without synovitis) at risk of developing RA, had elevated levels of let-7d-5p, miR-431-3p, miR-221-3p, miR-126-3p, and miR-24-3p, similar to RA patients [41].

Despite all the information published, there is limited acquaintance with the role of miRNAs in the preclinical phase of the disease [49]. Given that, we conclude that miRNAs are present in the circulation before the beginning of the disease. The involvement of differentially expressed cellular miRNA in different stages of RA progression and treatment (Table 1.), may offer information about which miRNAs have potential value as predictive and/or prognostic biomarkers in RA-cardiovascular complications.

### 2.2. Role of MiRNAs in RA-CVD Complications

Patients with RA develop a lot of cardiovascular complications, such as atherosclerosis, pericarditis and myocardial infarction. Unfortunately, some of these patients present at admission and at the time of diagnosis with already severe stages of cardiovascular pathologies [63]. Systemic pathological processes in RA, pro-inflammatory cytokines, namely IL-6 and TNF have been shown to increase atherogenesis [64]. Atherogenesis is one of the pillars of the development of subsequent cardiovascular consequences [65]. More and more data attributes an important role to epigenetic regulatory mechanisms, that influence not only the AID development but also cardiovascular complications onset and interrelations between these two [66,67]. Accordingly, irregularity in a plethora of miRNA and their associated functions have been described in systemic autoimmune diseases (SADs). It is, therefore, crucial to identify these early risks and initiate appropriate treatment as soon as possible [68].

#### 2.2.1. RA-Atherosclerosis

RA has been correlated with an increased risk of developing atherosclerosis [65]. There is a close connection between inflammation and atherosclerosis. The cytokines storm involved in the RA pathogenesis leads to different inflammatory-related processes [69]. Some miRNAs are well-known regulators of inflammatory pathways [68]. One of the miRNAs involved in both RA-CVD axis is miR-23a-5p. Its overexpression can inhibit cell proliferation, inflammation, and cell death in the RA synovial fibroblasts (RASFs) cells treated with TNF-α [62]. TNF-α is an inflammatory mediator that acts as a stimulator of the RASF proliferation [70], secretion of IL-6, matrix metalloproteinases (MMPs), prostaglandins, and of different effector molecules [71]. An increased plasma level of inflammatory cytokines and acute-phase reactants associated with prooxidative dyslipidemia, insulin resistance, and prothrombotic state, subsequently leads to endothelial dysfunction and arterial stiffness [72]. Further data shows that the miR-23a-5p/ATP-binding cassette transporter A1/G1 axis may promote plaque stability and macrophage-derived foam cell formation, which eventually inhibits atherosclerosis progression [73]. Moreover, in RA patients, Bao et al. [62] revealed that the increase of miR-23a-5p significantly inhibited the secretion of pro-inflammatory factors. Other miRNAs such as miR-146a [74], miR-766-3p [75] and miR-548a-3p [76] mediate the proliferation and inflammation of RA fibroblast-like synoviocytes by downregulating Toll-like receptors/the nuclear factor-κB (TLR4/NF-κB) signaling pathway. As TLR4/NF-κB signaling pathway is considered to be involved in the inflammation process leading to atherosclerosis, its downregulation may prevent the increased burden risk of the development of CVD. Also, recent reports are showing that a various miRNAs hold potential as markers of earlier stages of the atherosclerotic process [54,77].

Using the carotid intima-media thickness (cIMT) and the carotid plaque presence (cPP), Taverner and colleagues investigated the potential role as biomarkers of different miRNAs in CV prevention. They found out that miR-425-5p and miR-451 expression levels were able to significantly foretell pathological cIMT in men (*p* = 0.036) and women (*p* = 0.021), showing that a decrease in the expression of miR-451 in women is associated with lower arterial stiffness and that miR-425-5p in men is correlated with higher and lower values of subclinical arteriosclerosis [54]. Additionally, Ormseth et al. [77] noted that early evaluation of the circulating miR-30a-5p, and miR-125a-5p let-7c-5p, miR-126-5p, miR-4446-3p, miR-3168, miR-425-5p, miR-30e-5p, miR-126-3p improved the prognosis of high coronary artery calcium among patients with RA [77]. These studies may provide a basis for this field, in which specific miRNAs that can early predict the presence of coronary artery atherosclerosis, can be discovered and implemented in current clinical practice.

#### 2.2.2. RA-Myocardial Infarction

Myocardial infarction (MI) is one of the most important known worldwide causes of death among all cardiovascular diseases [78]. The increased risk of MI in a patient with RA is uncontested. RA patients with MI are more likely to have recurrent ischemia and mortality compared to healthy individuals [79,80]. In RA, miRNAs have been reported to play an important role in the pathogenesis of MI [81], by regulating different signaling pathways, especially apoptosis-related pathways such as (PI3K/AKT) and TLR4/NF-κB signaling pathway [82,83]. Notably, miR-23a-5p could be used not only as an independent risk factor for cardiovascular events having pro-apoptotic and anti-inflammatory effects in RA [84], but also, as a predictor for MI development [62]. In vivo study reported that reduced miR-23a-5p in the cardiomyocytes could weaken cardiac cell apoptosis, while in vitro miR-23a-5p-overexpression H9C2 cells induced decreased apoptosis at the same time with inhibition of the phosphorylation of PI3K/AKT [84]. Thus, miR-23a-5p may be a potential biomarker for the human systemic inflammatory response and also, may help improve cardiac efficiency during MI. In addition, different miRNA expression patterns of miR-1, miR-133a/b, miR- 150, and miR-186, miR-210, miR-320, miR-21, miR29, and miR-451 are potentially dysregulated in response to cardiac ischemia [85,86,87,88].

MiR-451 is highly expressed during myocardial infarction [89]. Its dysregulation of expression has been investigated in RA pathogenesis [35]. An excellent revision by Prajzlerová et al. [55] found dysregulation expression of miR-451 in peripheral blood mononuclear cells (PBMC). They found higher expression of miR-451 in PBMC from RA-risk individuals with arthralgia compared to HC (*p* ≤ 0.001) [55], which positively correlated with the visual analog scale (VAS) score of the patient’s global health (r = 0.484; *p* = 0.036). Based on these findings, the authors continued their research by investigating furthermore the potential of miR-451 target genes, their role in RA and how could it influence MI development. Researchers found that C-X-C Motif Chemokine Ligand 16 (CXCL16) expression is regulated by miR-545 [89,90] and by miR-451 [91]. CXCL16 is a chemokine with a potentially crucial role in RA and MI by chemoattraction of immune cells to the inflammatory locus [92]. These unexpected correlations probably indicate that both miR-451 and miR-545 have potential as biomarkers having functional roles in inflammation. More, the expression of the CXCL16 positively correlated with miR-451 expression secondary to the inflammation-induced expression of miR-451 [89], while miR-545 negatively regulated CXCL16 levels [90], inhibiting apoptosis in the MI cell, and diminishing CXCL16-induced injury on cardiomyocytes [90].

The association between miRNAs, signaling pathways, and genes may help to improve knowledge about the mechanisms underlying autoimmune diseases and their associated comorbidities.

#### 2.2.3. RA-Pericarditis

Besides affecting the joints, RA has different extraarticular manifestations (EAMs) such as rheumatoid nodules [1]. Patients with rheumatoid nodules are more often to develop severe EAMs as pericarditis or valvular heart diseases [93]. RA-pericarditis has a frequency of about (48.9%) and it is correlated to increased acute phase reactants, demonstrating a close and interdependent interrelationship between rheumatic disease activity and the evolution of pericarditis [94]. Patients with a severe form of the disease are more susceptible to developing pericarditis [95,96]. In the latest decades, various studies outlined the presence of miRNAs involved in the inflammatory activity in the pericardial liquid, and thus their implication in pericarditis development [97,98,99,100,101]. Beltrami et al. [102] demonstrated that let-7b-5p promotes angiogenesis, decreasing in the same time transforming growth factor beta receptor 1 (TGFBR1), while others noted its implications in pathologic processes such as oxidative stress [102,103]. Angiogenesis is a process involved both in pericardial inflammation and in synovial hyperplasia, by mobilization of the vascular endothelial growth factor (VEGF) and fibroblast growth factors (FGF), and by activation of proteolytic enzymes which results in impaired vascular permeability [104,105]. Taking its role in this process, we suppose that let 7b-5p is a promoter of angiogenesis that may be simultaneously involved in both inflammatory pericardial effusion and synovial fluid, and therefore can serve as a biomarker in the onset of CV-RA related disease. Lower serum levels of miR-146a define patients with early rheumatoid arthritis (ERA) [51]. A recent study revealed that rodents with induced constrictive pericarditis had early elevated levels of miR-146a [106].

These findings open new exciting knowledge about the miRNAs as biomarkers that hold potential in avoiding the onset of cardiovascular complications and highlights their diagnostic and prognostic power. Other microarrays and their role in CVD complications in RA are detailed in Table 2.

## 3. miRNAs-Therapeutic Approach

Operative therapeutic strategies in RA-CVD complications are limited by poor control of the basic disease. The development of bioengineering has allowed the more in-depth study of miRNAs, and as shown through this paper, emerging evidence point out the potential role of MiRNAs as therapeutic targets in RA and in RA-CVD complications [22,62,73,76,90].

A possible cellular target via miRNAs modulation envisioned recently, is represented by the RASFs. They are successors of synovial fibroblasts with an important role in RA progression by promoting the production of proinflammatory cytokine and enzymes that erode bone matrix leading at least to systemic inflammation [117]. In vitro studies revealed that inclusion of several miRNAs to RASFs may prevent the initiation of inflammatory processes [56]. For instance, miR-124a [118], miR-26b [119], miR-573 [120] are some of the miRNAs that reduced cytokine production in RASFs and prevented RASFs proliferation [17]. A study performed by Zhu et al. demonstrated that after transfection with the miR-140 mimics, in the miR-140-expressing cells the number of inflammatory cytokines decreased and apoptosis of RA-FLSs increased [50]. Furthermore, miRNA-140-5p was reported to act as a regulator of FLSs growth, invasion, apoptosis, and inflammation by targeting the signal transducer and activator of transcription 3 (STAT3). Evidence delineates a relation between miRNAs and the STAT3 pathway in RA-FLS progression [121].

Yang et al. revealed that miR-671-5p overexpression influences STAT3 protein levels in RA- FLSs. MiR-671-5p restricted the proliferation of RA-FLSs, while STAT3 declined the effects of miRNAs, those data suggesting the potential therapeutic approach of using miRNAs on the FLS and thus, prevent disease progression [122]. Besides, other miRNAs such as miR-653-5p influence the modulation of the inflammatory response in RA, by plug in cell migration, and invasion in FLs cells and by targeting Fibroblast Growth Factor 2 (FGF2) [123]. On the other side, overexpression of miR- 21 induces FLS proliferation in RA models through the NF-κB pathway [60], but can also, manage the TLR4/NF-κB pathway to reduce the release of inflammatory agents and diminishes myocardial cell injury in rats. Thus, miR- 21 can reduce myocardial cell apoptosis by mediating FGF1 to protect the myocardium [105]. We may conclude that increased expression of miR-21 could become an effective therapeutic solution in MI, however, in RA patients these miRNAs can lead to progression of the disease. Facing this dilemma, we suppose that either we can use miR-21 to stop the progression of the rheumatic disease and subsequent installation of CV complications, or in case MI already triggered targeting, miR21 may have a higher beneficial effect. However, further scientific data are needed to support these assumptions.

Another promising therapeutic approach, is illustrated by the capacity of miRNAs to inhibit angiogenesis via mediation of downregulation [102]. Via various MiRNAs we can avoid the onset of endothelial damage [124], as current knowledge displays the protective effects of miR-21 and miR149-5p on vascular endothelial cells [111,125]. Overexpression of miR-149-5p limited the vascular smooth muscle cells (VSMCs) proliferation, invasion, and migration [114], while miR-21 regulate vascular endothelial growth factor (VEGF) expression, and endothelial angiogenesis [60,102].

As, miRNAs are involved in the pathogenesis of RA onset and progression and in RA-CV complications, it is not unusual to see their therapeutic potential. The ideal approach using miRNAs, however, would be very complex and expensive. Nonetheless, hopefully in the near future additional studies may form a basis for this field.

## 4. miRNAs-Sequence Data

Having several roles as post-transcriptional regulators, miRNAs have become one of the most studied subjects in biotechnology [126]. Because of that, more experimental complex methods are needed for determining the sequence structure of miRNAs [127]. Thus, computational techniques were proposed to avoid the expensive costs of miRNAs research [128]. All databases were divided into seven categories [129] and formed the miRBase Sequence database, which is one of the main resource databases for miRNA sequence information [130]. Also, other frequently known used tools for miRNA detection are the DIANA microT, TargetScan, and miRanda [131]. Those bases provide a lot of information about expression profiling of specific miRNAs [18,132]. However, there are some advantages and disadvantages to these tools; for example, not all sequences reported in those databases have been experimentally validated, and also, there are many other miRNAs to be discovered as the exact number of miRNAs is far from being specified. According to that, there is an imperative need to improve and enlarge current sequence data and discover new array methods that can be used.

## 5. Conclusions

RA remains one of the most complex rheumatologic diseases with intriguing pathophysiology mainly represented by various immunological and inflammatory alterations that subsequently lead to systemic inflammation. In this manner, when untreated RA can progress over time to extraarticular manifestation, especially cardiovascular complications. RA-CVD not only have become a leading cause of death among these patients, but also, persist and raise challenges in clinical practice therapeutic strategies. For these reasons, the need for new biomarkers that can reveal the first signs of the disease and thus prevent its progression, and also, new additional therapeutic options are eagerly desired.

Not only, miRNAs have been correlated with RA onset and progression, but as seen, many scientific reports demonstrated the value of some miRNAs in monitoring disease activity and its evolution. Additionally, by regulating autoimmunity and inflammation miRNAs demonstrated their effectiveness as possible therapeutic targets (Figure 1). We can certainly only conclude at the end of this review, that miRNAs hold robust potential as future biomarkers and/or as novel therapeutic strategies complementary to conventional treatment. Nevertheless, in pursuance of putting a basis for a final miRNAs’ biomarker panel with low-cost detection method and for a new therapeutic miRNA non-invasive tool that could be used in general clinical practice, further larger studies are needed.

## Figures and Tables

**Figure 1 ijms-23-05254-f001:**
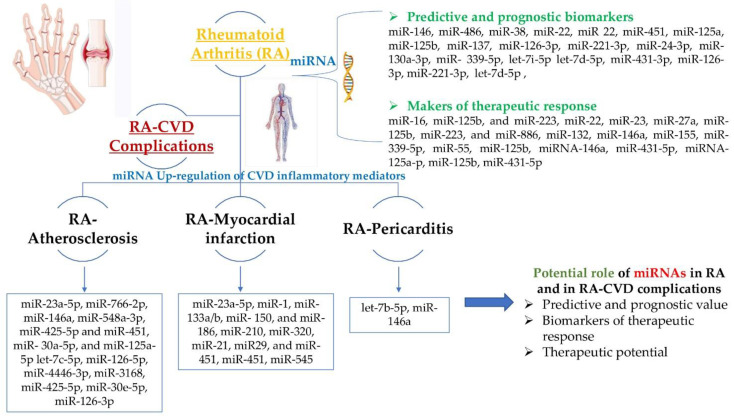
Summary of the involvement of specific miRNAs in RA and RA-CVD complications, and their potential value.

**Table 1 ijms-23-05254-t001:** miRNAs involvement in RA onset, progression, and disease treatment. Rheumatoid arthritis (RA); fibroblast-like synoviocyte (FLSs); healthy control (HC); Serine/Threonine Kinase 3 (AKT3); SMAD Family Member 3 (SMAD3); E2F Transcription Factor 8 (E2F8); tumor necrosis factor-alpha (TNFα); disease-modifying antirheumatic drugs (DMARDs); early rheumatoid arthritis (ERA); peripheral blood mononuclear cell (PBMC); interleukin-1 receptor-associated kinase 1 and 2 (IRAK1 and IRAK2); tumor necrosis factor receptor-associated factor 6 (TRAF6); Disease Activity Score (DAS28); c-reactive protein (CRP); erythrocyte sedimentation rate (ESR); early rheumatoid arthritis (ERA); anti-citrullinated protein/peptide antibody (ACPA); Member RAS Oncogene Family (RAB5A); C-X-C Motif Chemokine Ligand 16 (CXCL16); osteoarthritis (OA); matrix metalloproteinase (MMP-1, MMP-13); toll-like receptor (TLR4); normal controls (NC); Insulin Like Growth Factor 1 Receptor (IGF1R); Insulin Like Growth Factor Binding Protein 5 (IGFBP5); interleukin-1β (IL-1β); Matrix Metallopeptidase 13 (MMP13); Interleukin-6 (IL-16); lower (↓); raised (↑), no data (-).

miRNAs	Organ/Cell of RA	Association	Target Genes	Potential Roles	Ref.
**miR-149-5p**	FLSs	↓in the RA FLSs vs. HC *p* < 0.01; -↓pro-apoptotic activity	AKT3, Smad3, E2F8	-disease onset	[50]
**miR-146a-5p**	Serum Synovial fibroblasts	↑levels after TNFα/DMARDs treatment *p* < 0.05.	-	-therapy effectiveness	[37]
**miR-23a-3p**	Serum	therapy response-sensitivity of 62.5% and 57.1%, specificity of 86.4% and 90.2%.	-	prediction	[37]
**miR-146**	PBMC Synovial fluid Peripheral Blood	-↓ miR146a in ERA; - upregulated expression; ↑miR-146a ⇒ ↑inflammatory cytokine.	IRAK1, IRAK2, TRAF 6	prediction	[33,51,52]
**miR-223**	Serum Synovial fibroblasts	- correlated with DAS28; correlated with subcutaneous nodules -potential discriminator of RA *p* < 0.0001.	-	prognostic, prediction	[33,53]
**miR** **-425** **-5p**	Serum	- correlated negatively with ESR.	-	prognostic	[54]
**miR** **-451**	Synovial fluid	↓proliferation of synovial fibroblast; correlated with DAS28, ESR, CRP; ↑miR-451 in PBMC from ACPA-positive RA-risk individuals with arthralgia vs. HC.	CNEP3, Rab5a, CXCL16	prognostic, prediction	[28,54,55,56]
**miR-155**	PBMC	↑IL-17-, IL-1β- and LPS-stimulated IL-6; - suppression of SOCS1; ↑miR-155 ⇒↓SHIP-1 expression ⇒ ↓IL-6, Th17, TNF-α.	CCL2 release SHIP-1	prediction, prognostic	[57,58]
**miR-21**	FLS	associated with disease activity (r = 0.95, *p* < 0.01); promote NF- κB nuclear translocation FLS proliferation (*p*< 0.05).	NF- κB	prognostic	[59,60]
**miR-143**	Synovium	regulates MH7A cell proliferation; ↑miR-143-3p in RA than OA and the control (*p* < 0.01); ↓IL-1β, IL-6, IL-8, (MMP)-1 MMP-13 (*p* < 0.05).	IGF1R/IGFBP5	predictive	[61]
**miR-23a-5p**	Peripheral blood MH7A cells	-↓miR-23a-5p in RA compared NC; ↑miR-23a-5p inhibited TLR4 and p-NF-κBp65; promote cell apoptosis in TNF-α-treated RASFs	TLR4	prediction, prognostic	[62]

**Table 2 ijms-23-05254-t002:** miRNAs and their role in RA-CVD complications. Constrictive pericarditis (CP); rats group raised 8 weeks (CP-8W); 16 weeks group (CP-16W); normal group (N); Toll-like receptor 4 (TLR4); protein kinase B (Akt); the phosphatase and -tensin homolog (PTEN); signal Transducer And Activator Of Transcription 3 (STAT3); apoptosis Regulator (BCL2); bcl-2-like protein 4 (bax); fibroblast-like synovial cells (FLS); cardiac microvascular endothelial cells (CMECs); lipopolysaccharide (LPS); matrix metalloproteinase-3 (MMP-3); Interleukin-1 (IL-1 β); vascular endothelial growth factor (VEGF); matrix metalloproteinase (MMP-9); vascular smooth muscle cells (VSMCs); histone deacetylase 4 (HDAC4); nuclear factor erythroid 2-related factor 2 (Nrf2); sirtuin 2 (Sirt2); kelch-like enoyl-CoA hydratase-associated protein 1 (Keap1); heme oxygenase 1 (HO-1); pulse wave velocity (PWV); erythrocyte sedimentation rate (ESR); C-Reactive Protein (CRP); Disease Activity Score (DAS/DAS28); carotid intima-media thickness test (cIMT); the α-chemokine receptors (CXCR); lower (↓); raised (↑).

miRNAs	RA-CVD Comp.	Results	RA	Signaling Pathways	Site	Ref.
**miR-146**	Constrictive pericarditis	↑CP-8W group than in the N group and CP-16W group (*p* < 0.05), no difference between CP-16W and the N group (*p* > 0.05); ↓miR-146a ⇒ ↓MF markers.	-inhibit cellular inflammatory response	TLR4	Peripheral blood, tissues	[106]
**miR-21**	Myocardial infarction	↑angiogenesis via the PTEN-Akt pathway (*p* < 0.05); ↑ VEGF- protective effects, ↓ PTEN expression (*p* < 0.05); -activating STAT 3;	↑ STAT3 expression; -↑Bcl-2, ↓Bax expressions ⇒apoptosis	PTEN-Akt STAT3	FLS CMECs	[107,108,109,110]
**let-7c-5p**	Atherosclerosis	predictor of coronary calcium (c-statistic = 0.87 (95% CI 0.82, 0.93)); high-risk coronary calcium (c-statistic = 0.80 (95% CI 0.73, 0.86)).	negatively regulate levels of IL-1β, IL-6 and TNF-α expression.	Unknown	Synovial fibroblasts	[77]
**miR-221**	Myocardial infarction	myocardial ischemia-reperfusion by ↓PTEN; inhibit cardiomyocytes apoptosis.	↓the expression of cytokines, chemokine(*p* < 0.0); ↓ FLS stimulated by LPS; ↓ VEGF, MMP-3, MMP-9.	PTEN	Serum, FLS	[59,111]
**miR149-5p**	Endothelial damage, myocardial infarction	inhibit VSMCs proliferation; regulates VSMCs by targeting HDAC4.	↓ IL-1β, IL-6, and TNF-α ⇒ ↓ inflammation.	HDAC4	Serum, FLS	[112,113]
**miR-140-5p**	Hypertension	↑oxidative stress and ROS levels; ↓Nrf2, Sirt2 (*p* < 0.01), Keap1, HO-1.	inhibitory effect on FLS; ↓ pro-inflammatory cytokines; - ↓apoptotic rate;	STAT3	Serum, FLS	[50,114]
**miR-451**	Coronary artery disease Myocardial infarction	↓miR-451 ⇒↓cIMT in women (β = −0.05; *p* = 0.013); ↓miR-451⇒ ↓PWV (β = − 0.72; *p* = 0.035; ↑miR-451 -during myocardial infarction.	positively correlated DAS28 (r = 0.19; *p* = 0.006), ESR (r = 0.23; *p* = 0.001), CRP (r = 0.15; *p* = 0.033), fibrinogen (r = 0.28; *p* = 0.0001); ↓proliferation of synovial fibroblasts, production of cytokines.	Unknown	T cells, Neutrophils	[35,54,89,115]
**miR-425-5p**	Coronary artery disease	↓miR-425-5p ⇒↑cIMT in men (β = 0.072; *p* = 0.017); -not associated with PWV.	-prediction of high coronary artery calcium.	CXCR6	Serum	[54,77]
**miR-21-5p**	Heart failure	-inhibits cardiac fibroblast apoptosis; ↓miR-21 expression ⇒ ↑cardiac-hypertrophy, interstitial fibrosis.	increased cell invasion and decreased apoptosis in FLSs; -inhibited the expression of proapoptotic gene Bax.	PI3K/ AKT	FLS	[108,116]

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
