# Peer review of "MicroRNAs (miRNAs) in Cardiovascular Complications of Rheumatoid Arthritis (RA): What Is New?"

_ijms, 2022, doi:10.3390/ijms23095254_

Round 1

Reviewer 1 Report

Dear Authors,

This is interesting and qualified review that highlighted the specific role of miRNAs in 27 autoimmune processes, explicitly on their regulatory roles in the pathogenesis of RA, and its CV 28 consequences, their main role as novel biomarkers, and their possible role as therapeutic targets. 

There are no problems and contradiction in the review. However, please think to add references below:

  1. Autoimmune targeting of key components of RNA interference. (https://pubmed.ncbi.nlm.nih.gov/16684366/ ) This article is the oldest one to mention about the relationship between autoimmune/autoantibody and RNAi (microRNA). 
  2. MicroRNA-766-3p Contributes to Anti-Inflammatory Responses through the Indirect Inhibition of NF-κB Signaling. (https://pubmed.ncbi.nlm.nih.gov/30769772/) The article is about miR-766-3p that has anti-inflammatory effect.

Sincerely yours,

Author Response

Dear Reviewer,

Firstly, thank you on behalf of our team for your time and recommendations.

  1. We have included the reference suggested in the main text, as it was most suited; corresponding to ref [18] in the main text.
  2. Also, we have introduced information about miR-766-3p in a befitting context, its corresponding reference in the manuscript being: [73].

Reviewer 2 Report

This review paper is well-written, original, and looks valuable in this filed. I think it can be helpful for readers to know about role of miRNA in the cardiovascular complication in RA conditions. I got only few things.

  1. Summary figure should be required in conclusion section.
  2. Sequnce information of miRNA will be helpful.
  3. NF-B nuclear in table is not clear.
  4. TLR 3 or TLR-4 should unify as TLR3 and TLR4

Author Response

Dear Reviewer,

Firstly, thank you on behalf of our team for your time and recommendations.

  1. We have included in conclusion section a summary figure, as recommended. We hope that it describes the key findings of this review.

  1. As suggested, we have included a small paragraph in “5. miRNAs- sequence data” in which we briefly highlighted current knowledge about miRNA databases and known array methods.

  1. We have corrected in table 1 with the correct form “NF- κB nuclear” and correct information.

  1. We have unified from TLR-4/TLR 4 to TLR4.
